# Preparation of Succinoglycan Hydrogel Coordinated With Fe^3+^ Ions for Controlled Drug Delivery

**DOI:** 10.3390/polym12040977

**Published:** 2020-04-22

**Authors:** Yiluo Hu, Daham Jeong, Yohan Kim, Seonmok Kim, Seunho Jung

**Affiliations:** 1Department of Systems Biotechnology & Dept. of Bioscience and Biotechnology, Microbial Carbohydrate Resource Bank (MCRB), Center for Biotechnology Research in UBITA (CBRU), Konkuk University, Seoul 05029, Korea; lannyhu0806@hotmail.com (Y.H.); amir@konkuk.ac.kr (D.J.); shsks1@hanmail.net (Y.K.); gkdurk9999@naver.com (S.K.); 2Institute for Ubiquitous Information Technology and Applications (UBITA), Center for Biotechnology Research in UBITA (CBRU), Konkuk University, Seoul 05029, Korea

**Keywords:** succinoglycan, polysaccharide, metal coordination hydrogel, gel-sol conversion, drug release

## Abstract

Hydrogel materials with a gel-sol conversion due to external environmental changes have potential applications in a wide range of fields, including controlled drug delivery. Succinoglycans are anionic extracellular polysaccharides produced by various bacteria, including *Sinorhizobium* species, which have diverse applications. In this study, the rheological analysis confirmed that succinoglycan produced by *Sinorhizobium meliloti* Rm 1021 binds weakly to various metal ions, including Fe^2+^ cations, to maintain a sol form, and binds strongly to Fe^3+^ cations to maintain a gel form. The Fe^3+^-coordinated succinoglycan (Fe^3+^-SG) hydrogel was analyzed by attenuated total reflection Fourier transform infrared (ATR-FTIR) spectroscopy, circular dichroism (CD), and field-emission scanning electron microscopy (FE-SEM). Our results revealed that the Fe^3+^ cations that coordinated with succinoglycan were converted to Fe^2+^ by a reducing agent and visible light, promoting a gel-sol conversion. The Fe^3+^-SG hydrogel was then successfully used for controlled drug delivery based on gel-sol conversion in the presence of reducing agents and visible light. As succinoglycan is nontoxic, it is a potential material for controlled drug delivery.

## 1. Introduction

Hydrogels are “soft” materials that can absorb large amounts of water while maintaining their form in water long term [1]. Over the past few decades, hydrogels have been extensively applied in various fields, including tissue engineering [2], cosmetics [3], drug delivery systems [4], and sensors [5]. Hydrogels form a cross-linked polymeric three-dimensional network, which can be synthesized from one or more monomers or through a polymer [6]. In general, to form a three-dimensional network of hydrogels, the polymer is crosslinked by using a chemical crosslinking agent; alternatively, a crosslinking method is suitable for a polymer with physical or structural self-assembling properties [2,7]. In addition, hydrogels using inorganic crosslinking agents containing metal ions as well as organic crosslinking agents have been reported [8].

Hydrogels coordinated with Fe cations as crosslinkers have been well studied [9,10]. As transition metals, Fe cations can make covalent contributions in polymers that cannot be seen in ions, such as calcium, in coordination bonds [11]. Fe-coordination bonds function as a reversible and tunable hydrogel network in response to changes in the metal coordination environment, on the basis of the high electron affinity and transition between stable oxidation states. Polysaccharide-based hydrogel studies using these reversible Fe-ligand coordination bonds have been reported [12]. This is because most polysaccharides can form coordination bonds with metal ions, and depending on the structure, they can form strong complexes that form gels [13]. Polysaccharides are natural polymers obtained from renewable sources; they are low cost, highly soluble, and highly stable. In many studies, bacterial polysaccharides such as cellulose [14], alginate [15], xanthan gum [16], and gellan gum [17] were reported as basic hydrogel components. Xanthan gum, a microbial-derived polysaccharide, formed a hydrogel through coordination bonds with Fe^3+^ cations and showed a gel-sol conversion by a reducing agent [18]. In addition, the sacran-gel coordinated with Fe^3+^ cations that were gradually contracted by light irradiation energy [9], and the alginate hydrogel coordinated with Fe^3+^ cations that exhibited a photoresponsive activity [11]. The polysaccharides that form stimuli-responsive hydrogels by coordination with Fe^3+^ cations were anionic polysaccharides that can effectively bind Fe^3+^ cations. However, not all anionic polysaccharides and metal cations are coordinated to form hydrogels. In order for anionic polysaccharides to coordinate with metal cations to form hydrogels, the anionic polysaccharides must have a unique structure that can effectively sterically fit by combining with metal cations [10].

Succinoglycans are a type of exopolysaccharides (EPSs) secreted from *Rhizobium*, *Agrobacterium*, and species of soil microorganisms that play a crucial role in the development of the root nodule symbiosis between the bacteria and legumes of Alfalfa [19]. Succinoglycans are known for their viscosifying activity [20], emulsification property [12], and pseudo-plasticizing activity [12], and can be used to stabilize brine solutions, as a fluid-loss controlling agent, and as a cosmetic additive [12]. With a unique helical structure, succinoglycan, as an anionic EPS, consists of octasaccharide repeating units containing one galactose residue and seven glucose residues with β-1,3, β-1,4, and β-1,6-linked subunits; each unit is modified with one acetyl group, one or two succinyl groups, and one pyruvate group [21]. Low molecular weight succinoglycans can be chelated with Fe^2+^ to provide antioxidant activity through an anti-Fenton reaction, thereby effectively controlling Fe biochemistry [22]. However, there have been no reports of hydrogel formation through the coordination of succinoglycan with Fe^3+^ cations.

In this study, we isolated succinoglycan from *Sinorhizobium meliloti* Rm 1021, and, to our knowledge, hereby present the first evidence that it effectively coordinates with Fe^3+^ cations to form a hydrogel. Fe^3+^-coordinated succinoglycan (Fe^3+^-SG) hydrogels were investigated by various methods, including rheometry, attenuated total reflection Fourier transform infrared (ATR-FTIR) spectroscopy, circular dichroism (CD) spectropolarimetry, and field-emission scanning electron microscopy (FE-SEM). Fe^3+^-SG hydrogels were able to control the release of drugs through the gel-sol conversion, depending on the photoreductant concentration and a reducing agent capable of reducing Fe^3+^ to Fe^2+^.

## 2. Materials and Methods 

### 2.1. Chemicals

Congo red and 1,10-phenanthroline were purchased from Sigma-Aldrich Chemicals Co., St. Louis, MO, USA. A sodium lactate 50% solution was obtained from Duksan Pure Chemicals Co., Ltd. Ascorbic acid was purchased from the Beijing Chemical Works Reagents Company, Beijing, China. Irradiation with visible light was performed by using a 405 nm laser.

### 2.2. Isolation and Purification of Succinoglycan

Succinoglycan was produced from *S. meliloti* Rm 1021 supplied by the Microbial Carbohydrate Resource Bank (MCRB) of Konkuk University, Korea. The bacteria were grown in a production medium at 25 °C for 14 days. The production medium comprised of d-mannitol (50 g/L), glutamic acid (7.5 g/L), K_2_HPO_4_ (15 g/L), KH_2_PO_4_ (15 g/L), MgSO_4_.7H_2_O (1 g/L), and CaCl_2_ (0.2 g/L), and the pH was adjusted to 7.0. Microbes were then removed by centrifugation at 8000 g for 15 min; three volumes of ethanol were added to precipitate the supernatant, and the precipitate was collected using filter paper. Subsequently, the collected precipitate was purified by dialysis (MWCO 12–14 kDa membrane) with distilled water (DW) for three days and lyophilized after collection to obtain the purified succinoglycan [23].

### 2.3. Rheological Measurements

The rheological experiments were performed by using a DHR-2 rheometer (Thermo Fisher Scientific, Waltham, MA, USA). To demonstrate the ability to form hydrogels through the coordination of succinoglycan with metal ions, a 1.0% (*w*/*v*) succinoglycan solution was first prepared and then mixed with a 40 mM metal ion solution in a 1:1 volume ratio. The storage modulus (G’) and loss modulus (G”) of hydrogels was measured at 25 °C by applying a 1.0% strain between 0.1 and 100 rad/s.

### 2.4. Attenuated Total Reflection-Fourier Transform Infrared (ATR-FTIR) Spectroscopy

The ATR-FTIR spectroscopy analysis was conducted by using an ATR-FTIR spectrometer (Spectrum Two FTIR, Perkin Elmer, Waltham, Massachusetts, USA) equipped with a PIKE MIRacle ATR accessory. According to the conditions mentioned in the rheological studies, a solution of Fe^3+^ and Fe^2+^ cations was mixed with a 1.0% (*w*/*v*) succinoglycan solution, respectively. Succinoglycan mixed with Fe^2+^ cations in the solution formed a sol; in contrast, the mixture with Fe^3+^ cations formed a gel. Three samples, including the succinoglycan, were frozen in the deep freezer, and then lyophilized. Spectra were collected in transmission mode in the range from 4000 to 500 cm^−1^ at a resolution of 0.5 cm^−1^ using 10 scans.

### 2.5. Circular Dichroism (CD) Spectropolarimetry

The CD spectra of samples in the wavelength range from 190 to 260 nm were recorded by using a JASCO J-810 spectropolarimeter. All samples contained 0.05% (*w*/*v*) succinoglycan and 0.02 mM Fe^2+^ and Fe^3+^ cations in DW. Experiments were performed in a 0.1 cm path length cuvette at 25 °C and expressed, on average, five scans. The response time and bandwidth were 2 s and 0.2 nm, respectively.

### 2.6. Preparation of Fe^3+^-SG Hydrogel Beads

Succinoglycan 1.0% (w/v) was dissolved in DW and placed in individual glass vials. The succinoglycan solutions were treated with different concentrations of Fe^2+^ or Fe^3+^ solutions (0, 3, 6, 15, 30 mM), and the resulting mixtures were stirred gently at room temperature. The vials were then turned upside down to determine whether gelation had occurred or not. In the vial tests, Fe^3+^-SG hydrogel beads were prepared, as follows. The prepared 1% (*w*/*v*) succinoglycan solution was added dropwise to an aqueous 30 mM Fe^3+^ solution through a 10 µL pipette tip. The beads were kept in a Fe^3+^ solution for 15 min to ensure that sufficient coordination occurred. Subsequently, the Fe^3+^-SG hydrogel beads were washed three times with DW to remove the excess Fe^3+^ cations. Next, the prepared Fe^3+^-SG hydrogel beads were analyzed by FE-SEM, UV–Vis spectroscopy, Congo red release, and cytotoxicity.

### 2.7. Field Emission-Scanning Electron Microscopy (FE-SEM) Analysis 

The Fe^3+^-SG hydrogel beads produced were rinsed three times in DW and were lyophilized overnight. The beads were cut in half to analyze their internal structures, which were visualized by using FE-SEM. The beads were coated with a platinum layer at 30 W for 30 s in a vacuum prior to the FE-SEM analysis.

### 2.8. Spectrophotometric Detection of the Reduction of Fe^3+^ to Fe^2+^

In this study, the reduced Fe^2+^ cations were measured by UV–Vis spectrometry, as Fe^2+^ cations were reacted with 1,10-phenanthroline to form a stable Fe^2+^-1,10-phenanthroline complex (Fe^2+^-phen), showing a maximum absorption at 510 nm [24]. Based on this, the concentration of reduced Fe^2 +^ cations was measured under two conditions. First, Fe^3+^-SG hydrogel beads (20 mg beads) were stored in 5 mL vials. Next, 2 mL of a 10 mM ascorbic acid solution was added to the vial and gently shaken. Every 10 min, stored aliquots of the suspension of Fe^3+^-SG hydrogel beads were collected and equilibrated with 1,10-phenanthroline for 2 min, diluted, and the UV–Vis absorption of the resultant beads was measured in the range of 400–600 nm. In the second condition, the prepared Fe^3+^-SG hydrogel beads were stored in 2 mL of a 10 mM sodium lactate solution, then Fe^3+^-SG hydrogel beads were irradiated by a diffuse 405 nm laser beam. Irradiation tests were conducted at 10 min intervals to record the UV–Vis spectra.

### 2.9. Congo Red Loading and Release

For Congo red loading studies, 50 µL of a Congo red solution (1 mg/mL) was mixed with 450 µL of a 1% (w/v) succinoglycan solution for 30 min. The final concentration of Congo red was adjusted to 0.1 mg/mL [11]. The Congo red loading amounts were determined by the extraction method [25]. Lyophilized Congo red-loaded Fe^3+^-SG hydrogel beads were ground then dissolved in a PBS buffer (pH = 7.4). The suspensions were stirred for 2 h, subsequently centrifuged at 13,500 g for 5 min. The extracted weight of Congo red was measured in the same manner as for the Congo red release studies. The Congo red loading amount was calculated as follows:$$\mathrm{Loading}\;\mathrm{amount}\;\mathrm{of}\;\mathrm{the}\;\mathrm{Congo}\;\mathrm{red}=\frac{\mathrm{Extract}\;\mathrm{weight}\;\mathrm{of}\;\mathrm{Congo}\;\mathrm{red}}{\mathrm{Weight}\;\mathrm{of}\;\mathrm{the}\;\mathrm{hydrogel}\;\mathrm{beads}}$$

The encapsulation efficiency of Congo red in Fe^3+^-SG beads was estimated as follows [26,27]:$$\mathrm{Encapsulation}\;\mathrm{efficiency}\;\left(\%\right)=\frac{\mathrm{Loading}\;\mathrm{amount}\;\mathrm{of}\;\mathrm{Congo}\;\mathrm{red}}{\mathrm{Weight}\;\mathrm{of}\;\mathrm{the}\;\mathrm{theoretical}\;\mathrm{Congo}\;\mathrm{red}}\times 100$$

The theoretical loading amount of Congo red in Fe^3+^-SG beads was 11.11 mg/g beads. Congo red-loaded Fe^3+^-SG hydrogel beads (50 mg) in a 3 mL PBS buffer solution (pH 7.4) were treated with either ascorbic acid as a chemical reductant or visible light as a photoreductant. At specific time intervals, the release of Congo red was determined at 498 nm. The cumulative release of Congo red was calculated from the following formula [25]:$$\mathrm{Cumulative}\;\mathrm{amount}\;\mathrm{of}\;\mathrm{the}\;\mathrm{drug}=C_{n}V+\overset{i=n-1}{\underset{i-1}{\sum }}C_{i}V_{i}$$
where *V* is the release of the medium volume, $V_{i}$ is the sampling volume, and $C_{n}$ and $C_{i}$ are the Congo red concentrations in the release medium and the aliquots.

### 2.10. Cytotoxicity Study

We used the human embryonic kidney 293 (HEK293) cell line purchased from the Bank of Korea Cell Line (Seoul, Korea) [28]. The cells were maintained in high-glucose Dulbecco’s modified Eagle’s medium (DMEM; Hyclone, Logan, UT, USA) supplemented with a 10% fetal bovine serum (FBS) and 1% antibiotics (100 U/mL penicillin and 100 g/mL streptomycin) at 37 °C (humidified, 5% CO_2_) [29]. In the direct cytotoxicity experiments, we treated 5 mg/mL of lyophilized succinoglycan, Fe^2+^-SG sol, and Fe^3+^-SG hydrogel beads in 96-well plates containing 5 × 10^3^ cells per well, respectively. In addition, the indirect extract cytotoxicity test was performed as described in the ISO 10993-5:2009—Biological evaluation of medical devices, part 5: Tests for in vitro cytotoxicity, by indirect contact [29]. Indirect extract samples were prepared as follows: 30 mg of Fe^3+^-SG hydrogel beads were immersed in a 10 mL DMEM and then incubated for 12 h at 25 °C in the dark. Thereafter, the cells were placed in a 96-well plate (Costar, Cambridge, MA, USA) containing 5000 cells/well, treated with a prepared stock solution and indirect extract, which incubated for 24 h. After 24 h, cells were washed with a PBS buffer and the WST-1 reagent (EZ-Cytox; Daeil Lab Service Co. Ltd., Seoul, Korea) was added to each well [29]. After incubation for 4 h at 37 °C, cell viability was determined using a SpectraMax 190 microplate reader (Molecular Devices, Corp. CA, USA) at 450 nm. All experiments were conducted in triplicate.

## 3. Results and Discussion

### 3.1. Rheological Measurements

The rheological changes in the coordination bonds with various metal ions of succinoglycan were investigated. Succinoglycan isolated from *S. meliloti* was characterized by ^1^H NMR and gel permeation chromatography (GPC). The chemical structure is shown in Figure 1. It consists of one galactose and seven glucose residues, with a pyruvate group linked to the terminal glucose residue of the side chain [30]. As shown in Appendix A, in the ^1^H NMR data of succinoglycan produced by *S. meliloti* Rm 1021, the peaks with chemical shifts at 1.43 ppm represented methyl protons of the 1-carboxyethylidene (pyruvate); the peaks with shifts at 2.08 ppm represented methyl protons of acetate groups; the broad peak at 2.64 ppm represented the methylene protons of the succinate groups; and the complex region from 3.3 to 4.0 ppm indicated the protons of the carbohydrate backbone constituents. ^1^H NMR spectroscopy showed the fully acetylated succinoglycan containing approximately one or two succinate, one acetate, and one pyruvate groups. The average molecular weight of the succinoglycan was 354,839 g/mol (Appendix A). Figure 2 shows the storage modulus (G’) and loss modulus (G”) of the system measured through angular frequency (0.1–100 rad/s) [31]. The storage modulus (G’) describes the elastic properties and loss modulus (G”) characterizes viscous properties [32]. As shown in Figure 2a, the storage modulus of succinoglycan was always lower than the loss modulus, indicating that succinoglycan was fluidic. When succinoglycan was mixed with the metal ion solution (Na^+^, K^+^, Ca^2+^, Fe^2+^, Al^3+^, and Fe^3+^), all the same phenomena as the original succinoglycan solution alone were observed, except for the Fe^3+^ cation, in which the succinoglycan and Fe^3+^ cation mixture solution formed a yellowish hydrogel (Appendix A). As shown in Figure 2b, similar to succinoglycan, Fe^2+^-coordinated succinoglycan (Fe^2+^-SG) did not show any significant difference, with the storage modulus always lower than the loss modulus, indicating that Fe^2+^-SG was a liquid-like sol. Both G’ and G” were sensitive to the angular frequency and did not cross over each other in the tested range (Figure 2c). Fe, as a general transition metal ion, could create a coordinate covalent bond to several ligands, including nonbonding electrons [33]. The Fe^2+^ cation is a kind of “soft” metal ion that binds with neutral ligands such as nitrogen and sulfur atoms; in contrast, the Fe^3+^ cation is a “hard” metal ion that can coordinate with negatively charged ligands such as carboxylate, phenolate, and hydroxamate groups [34]. In the frequency range of 1–100 rad/s, the G’ value of a succinoglycan was 1.13 Pa and the G” value was 2.41 Pa. However, after the Fe^3+^ cations were coordinated with succinoglycan, G’ and G” increased to 64460.60 and 6255.06 Pa, respectively. These observations implied that when a dynamic load is applied, the Fe^3+^-SG hydrogels have a strong physical gel structure, indicating a highly elastic response with a comparatively small dissipation in energy [35,36]. In addition, the slope of G’ of the Fe^3+^-SG hydrogel is close to zero, which indicated that the storage modulus and angular frequency were associated with the dynamics of the hydrogel network [37]. Succinolgycan also coordinated with Fe^2 +^, thus, high concentrations of Fe^2+^ ions would be expected to increase the storage modulus and became a gel. In that view point, increasing the concentration of Fe^2+^ or Fe^3+^ ions would increase the storage modulus of the succinoglycan solution, such as Ca^2+^-alginate gel [38,39]. However, since the aqueous 1% succinoglycan was not completely dispersed in water, the addition of a high concentration of iron ions would form an uneven gel. These results clearly showed that succinoglycan coordinated with Fe^3+^ cations to form a strong hydrogel, whereas coordination with Fe^2+^ cations formed a fluid-liquid sol.

### 3.2. Attenuated Total Reflection-Fourier Transform Infrared (ATR-FTIR) Spectra Analysis

The coordination of Fe^3+^ cations with succinoglycan was identified by the magnitude of change in the absorption peak by using ATR-FTIR spectroscopy. The ATR-FTIR analysis is a superior tool to determine the altered structure of metal ion-coordinated polysaccharides. [40]. As shown in Figure 3a, the broad absorption peak at 3351 cm^−1^ indicated the existence of the stretching vibration of hydroxyl groups, while the absorption peak at 1724 cm^−1^ was attributed to the C=O stretching carbonyl ester of the acetate group [20,41,42]. The absorption peaks at 1626 and 1379 cm^−1^ were a result of the asymmetrical C=O stretching vibration of the succinate and acetate functional groups and symmetrical stretching vibration of the carboxylate –COO^−^ group from acid residues, respectively [20,41]. In addition, the absorption peak at 1045 cm^−1^ is a characteristic peak. The absorption peaks at 1724 and 1379 cm^−1^ were indicative of a symmetrical C–O stretching vibration from the sugar backbone [42]. When succinoglycan was coordinated with Fe^2+^ cations, the peak at 1724, 1379, and 1045 cm^−1^ were shifted to 1737, 1366, and 1070 cm^−1^, respectively, whereas no significant change at the absorption peak of 1626 cm^−1^ was observed (Figure 2b). These results showed that, in the case of succinoglycan coordination with Fe^2+^, the absorption peak 1724 cm^−1^ was shifted to 1737 cm^−1^ due to the electrostatic interaction between the O atom in C=O carbonyl of the acetate group and Fe^2+^ cation [43]. When succinoglycan was coordinated with Fe^3+^ cations, each of the absorption peaks at 1724, 1626, 1379, and 1045 cm^−1^ was shifted to 1699, 1622, 1397, and 1034 cm^−1^, respectively (Figure 3c), which indicated that Fe^3+^ coordination occurred between the carboxyl group of succinoglycan [44,45]. As shown in Figure 3a, the –OH stretching bands appeared at 3366 cm^−1^ for the Fe^2+^-SG sol (Figure 3b) and, 3371 cm^−1^ for the Fe^3+^-SG hydrogel (Figure 3c). These results exhibited that compared with the original succinoglycan (Figure 3a), the –OH stretching bands of succinoglycan that coordinated with the Fe^3+^ cations was shifted to higher frequencies, which proved that the metal coordination interaction occurred between the –OH group of succinoglycan molecules and iron ions [46]. Hence, these results indicated that Fe^3+^ cations were strongly coordinated with functional groups including the OH group of succinoglycan, unlike Fe^2+^ cations.

### 3.3. Circular Dichroism (CD) Analysis 

The CD spectrum of polysaccharides was used as a tool to suggest the indicator of a secondary structure [47]. In the case of native succinoglycan (Figure 4), a characteristic spectrum with a negative band centered at approximately 200 nm was observed [48]. This band corresponded to the n → π * transition by the carboxyl and carboxylates of pyruvate and succinate [49]. As Fe^2+^ was added to succinoglycan, the n → π* negative transition band due to the carboxyl and carboxylate of succinoglycan weakened (Figure 4; inset). This decrease in transition intensity was expected owing to the binding of Fe^2+^ to the carboxyl groups responsible for the transition, and the decrease in the intensity of transition may have been due to the Fe^2+^ complexation of succinoglycan, similar to previously reported studies [50]. In contrast, Fe^3+^-SG showed the characteristic CD spectra with sharp positive bands, both at ~195 and 202–208 nm, and sharp negative bands at ~198 nm. The emergence of these new bands could be attributed to the charge transfer interactions between Fe^3+^ cations and the carboxyl group [51]. As shown in Figure 1, succinoglycan has a backbone with regular side groups formed by four β-o-linked glycopyranose residues where the main chain contains two consecutive β-(1→6) glycoside bonds, one of which links the side chain to the main chain. This special connection may give flexibility to the side arms of succinoglycan. The fixed charges of pyruvate and succinate linked to this side chain strengthen it, and the uncharged backbone is somewhat strengthened by the entanglement of the residues of glycofuranose and galtopyranose, which create a regular and helical structure [52]. Owing to the spatial demands of charged bulky side chains, succinoglycan is likely to be a single helix with partial lateral aggregation [48]. Structural change in the succinoglycan side chain with a low ionic strength by complexation of the Fe^2+^ of succinoglycan did not appear to be as large as shown in the CD spectrum, and this result was similar to the previously reported Na^+^ ion complexation with succinoglycan [48]. However, the strong binding of Fe^3+^ cations to succinoglycan in an aqueous solution greatly reduced the flexibility of the side chains, and could also be expected to be complexed with the hydroxyl groups of a rigid backbone, as confirmed by the ATR-FTIR results. The anionic polysaccharide xanthan is known to undergo significant changes in physicochemical behavior owing to the removal of charged groups [53]. These results confirmed that the secondary structure of succinoglycan was changed dramatically by coordination with Fe^3+^ cations to form a gel. 

### 3.4. Fe^3+^-SG Hydrogel Bead Preparation

Bead materials can be integrated into microfluidic chips, optical fibers, microwells, and tips, and the beads are a useful material in various areas, such as disease diagnostics, and biological and chemical analyses. In this study, the beads were fabricated for an examination of the behavior of the gel-sol conversion of the Fe^3+^-SG hydrogel by using a reducing agent. The concentration at which the Fe^3+^-SG hydrogel was formed was checked before making the beads. In Fe^3+^-polysaccharide hydrogel systems, the concentration of the Fe cation is an important point in the physical state of the sample and can also determine whether it can form a uniform solid gel. The progress of the gelation was monitored by mixing different concentrations of Fe cations with succinoglycan aliquots (Figure 5). As shown in Figure 5a, the gelation started to occur when the concentration of the solution of Fe^3+^ cations was a 6 mM solution. Gelation was observed even when the Fe^3+^ cation concentration was 15 mM, but it was not able to maintain a specific form of the hydrogel. When the Fe^3+^ cation concentration was increased to 30 mM, it was confirmed that a strong gel could be formed. However, in the presence of Fe^2+^ (Figure 5b), the succinoglycan solution remained a free-flowing solution, even when the Fe^2+^ concentration was increased to 30 mM. Based on this observation, hydrogel beads prepared by the dropwise addition of a 1% (*w*/*v*) succinoglycan solution to a 30 mM Fe^3+^ cation solution were used. The diameters of Fe^3+^-SG hydrogel beads were measured using an optical microscope (magnification 10×) [54]. To determine the average bead size, measurements were conducted for randomly 10 beads. Figure 5c revealed that the beads are of spherical shape, with a relatively uniform size of ~2 mm. 

### 3.5. FE-SEM Micrograph Analysis

The FE-SEM analysis was conducted to observe the morphology and internal structure of the hydrogel. FE-SEM images of succinoglycan, Fe^3+^-SG hydrogel beads, the bead surface, and a cross-section of the bead are shown in Figure 6. As presented in Figure 6a, the original succinoglycan had a planar shape, a similar morphology to the low-molecular-weight succinoglycan reported by Kim et al. [28]. The round three-dimensional shape of Fe^3+^-SG hydrogel beads from the coordinating succinoglycan with Fe^3+^ cations was observed, as shown in Figure 6b, with a “zoomed in” detail of the surface of the Fe^3+^-SG hydrogel bead shown in Figure 6c. Analysis of the detailed morphology of the Fe^3+^-SG hydrogel bead outer surface showed a rough and corrugated structure. This rough, dense surface appearance was because of the strong binding of succinoglycans with Fe^3+^ ions on the surface of the beads when the succinoglycan is added into the Fe^3+^ solution to form a bead [55]. However, the cross-section of the hydrogel beads showed many reticular pores, unlike the surface, which means that the crosslinking by the Fe^3+^ cation was effective, even in the beads. The cross-section of the Congo red-loaded Fe^3+^-SG hydrogel beads still appeared as pores, and after the drug was released, it became a sol and the pores disappeared (Appendix A). These pores of the Fe^3+^-SG hydrogel beads would be expected to effectively load chemical compounds such as drugs.

### 3.6. Redox-Responsive Fe^3+^-SG Hydrogel Beads

Stimuli-responsive hydrogels can change their structure and properties in response to changes in their environment [56]. Moreover, some specific hydrogels can respond to different types of stimuli such as temperature, pH, solvent condition, and light irradiation, which affect a multi-responsive hydrogel system [31,57]. In particular, gel-sol conversion hydrogels based on the transition of Fe ions have attracted attention because the latter can transition between two stable oxidation states [18]. The chemical reduction of Fe^3+^ to Fe^2+^ in alginate coordinated-Fe^3+^ hydrogels produces calcium cross-linked alginate hydrogels in the presence of Ca^2+^ salts and ascorbic acid [58]. Ascorbic acid is often used for the reduction of Fe ions owing to its strong reductive potential and biocompatibility [18]. Ascorbic acid was found in small amounts in the human plasma [59], but is a substance that can induce the gel-sol transition of hydrogel in the human body by administration at high concentrations [18]. To effectively study the gel-sol conversion for Fe^3+^-SG hydrogel beads under different conditions in this study, the Fe^2+^-phen method was used to determine the reduction of Fe^3+^ to Fe^2+^ in coordinated succinoglycan and was found to be time-dependent. As shown in Figure 7a, the absorbance at 510 nm was increased as the concentration of Fe^3+^ coordinated with succinoglycan in the ascorbic acid solution decreased. After 140 min, an almost complete reduction of Fe^3+^ to Fe^2+^ was observed, as presented in Figure 7a (inset). It has been observed that the Fe^3+^-SG hydrogel beads can cause gradual conversion from gel to sol in the presence of ascorbic acid. The Fe^3+^-SG hydrogel beads responded not only to the stimulus of the reducing agent, but were also affected by light. As shown in Figure 7b, when the sodium lactate solution containing the hydrogel beads was irradiated with a 405 nm laser, the absorbance of the Fe^2+^-phen composite was observed to be increased at 510 nm [24]. Sodium lactate is known as a sacrificial photoreductant to increase the photoreduction rates owing to the lactic acid efficient bidentate binding of Fe^2+^ cations [34]. Compared with Figure 7a, the graph showed that the absorbance was affected by light within the first 60 min; however, no significant change was observed after that. In other words, this indicated that the Fe^3+^-SG hydrogel has a greater transfer effect by the reducing agent than by light. Thus, the Fe^3+^-SG hydrogel, as a redox-responsive hydrogel, can be used as a medical material for controlled drug delivery [60].

### 3.7. Congo Red Release

Congo red is a good drug model that can be used in drug release experiments because it is photo stable and nonreactive [11]. As an anionic diazoic dye, Congo red has a planar aromatic structure and dissolves in many solvents [61]. The behavior of Congo red in the solution is similar to aromatic drugs that produce red colloidal fluorescent solutions in an aqueous medium prepared due to the stacking mechanism. Congo red has also been used to detect fibrillar proteins useful for histological studies of some neurodegenerative pathologies such as Alzheimer’s, Creutzfeldt-Jakob, Huntington’s, and Parkinson’s diseases [62]. In this study, we prepared Fe^3+^-SG hydrogel beads by loading Congo red in the succinoglycan solution. The Congo red loading amount of Fe^3+^-SG hydrogel beads was 10 mg/g beads as confirmed by the extraction method. The Congo red encapsulation efficiency in Fe^3+^-SG beads were determined to be about 90% against a theoretical loading amount. This high encapsulation efficiency is expected to reduce the loss due to the diffusion of Congo red because the Fe^3+^ ion and the succinoglycan was rapidly cross-linked. According to the biological environment of humans with pH 7.4, we chose the PBS buffer (pH = 7.4) as a Congo red release media. The cumulative release of Congo red from Fe^3+^-SG hydrogel beads in buffers of various ascorbic acid concentrations is shown in Figure 8a. The Fe^3+^-SG hydrogel beads were released for only 6% of Congo red over 150 min in the PBS buffer without any reducing agent. This is because on the surface of the hydrogel beads, as shown in the FE-SEM results, succinoglycans are strongly bound to Fe^3+^ cations, which interferes with the release of the drug molecules (Figure 6c). In addition, it was confirmed that the release rate of Congo red in the beads was sensitive to the concentration of external ascorbic acid. For 10 mM of ascorbic acid, the amount released over 150 min was 45%; however, for 30 mM, a complete release occurred within that period. Figure 8b presents the release tendency of Congo red under the 405 nm of visible light showing the similar release characteristics to that of the ascorbic acid. The release of Congo red increased with the concentration of sodium lactate, a photoreductant, but not as fast as the case of ascorbic acid. This result was consistent with the previous Fe^2+^-phen complex study that the reduction of Fe^3+^ by light was less effective than the reduction by the reducing agent (Figure 7). To study the effect of Fe^3+^-SG hydrogel beads on cargo release stimuli-responsive properties, the Congo red-loaded Fe^3+^-SG hydrogel beads were prestored in the PBS buffer, which, after 60 min, were subjected to ascorbic acid and visible light, with the results shown in Figure 8c,d. As a result, Congo red was not released during the previous 60 min, but the loaded Congo red was released when stimuli, such as reducing agents or visible light, were present. Hence, the Fe^3+^ coordinated with succinoglycan has the potential as an effective controlled drug delivery system.

### 3.8. Cytotoxicity Tests

Since hydrogels have been widely used as drug carriers, in tissue engineering, etc., the cytotoxicity of hydrogels has a great impact on further research. A previous study has confirmed that succinoglycan has low toxicity to organisms [28]. Cell viability was measured by the WST-1 assay method; the results of cytotoxicity assays are shown in Figure 9. Compared with the control, the cell viability of succinoglycan, dried Fe^3+^-SG hydrogel beads, and indirect extract was 90%, 95%, and 91%, respectively. DMSO-treated cells were used as a positive control for cytotoxicity, and less than 20% of cells are viable under these conditions. As shown in Appendix A, regardless of the valence of the iron, the presence of the succinoglycan coordination with Fe^2+^ also did not affect cell viability. These results confirmed that Fe^3+^-SG hydrogel beads favored biocompatibility and cell viability, suggesting that the Fe^3+^-SG hydrogel has great potential as a future biomedical material. 

## 4. Conclusions

Succinoglycan isolated from *S. meliloti* Rm 1021 was subjected to structural analysis using ^1^H NMR spectroscopy and GPC. The rheological analysis suggested that when succinoglycans were mixed with Fe^2+^ cations, they formed a fluid solution; however, in the presence of Fe^3 +^ cations, strong gels could be formed through coordination bonds. CD spectra and ATR-FTIR confirmed that Fe^3+^ cations can establish coordination bonds with succinoglycan groups such as hydroxyl, succinate, pyruvate, and acetate to form a strong Fe^3+^-SG hydrogel. We confirmed that the Fe^3+^-SG hydrogel beads can undergo gel-sol conversion when stimulated with a reducing agent and visible light. The Fe^3+^-SG hydrogel has internal pores to load the drug, and because of the strong coordination between succinoglycan and Fe^3+^ cations, no pore formation occurs on the surface. Thus, the loaded drug is released gradually in the PBS buffer. However, in the presence of a reducing agent, the drug could be released rapidly. In addition, the nontoxicity of Fe^3+^-SG hydrogel beads indicated that these hydrogel beads exhibited a superior biological safety. These results reveal that the Fe^3+^-SG hydrogel may serve as a potential stimuli-responsive release system.

## Figures and Tables

**Figure 1 polymers-12-00977-f001:**
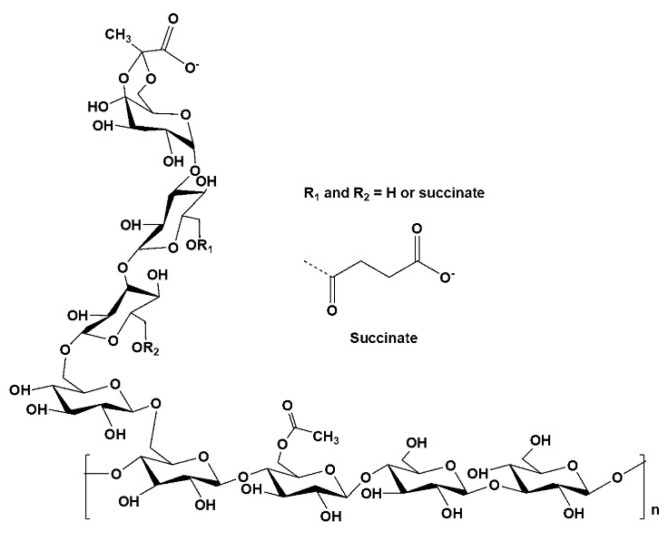
Structure of the succinoglycan repeating unit from *S. meliloti* Rm 1021.

**Figure 2 polymers-12-00977-f002:**
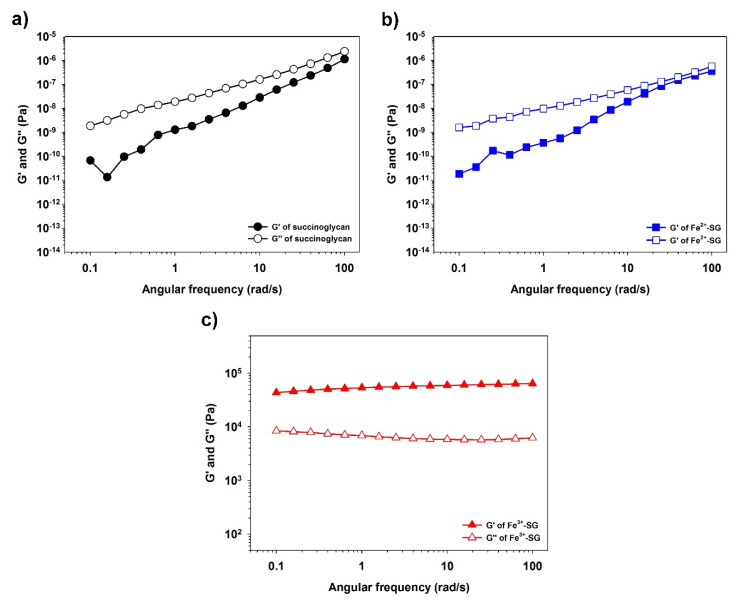
The rheological analysis of succinoglycan (**a**), Fe^2+^-coordinated succinoglycan (Fe^2+^-SG) sol (**b**), and Fe^3+^-SG hydrogel (**c**).

**Figure 3 polymers-12-00977-f003:**
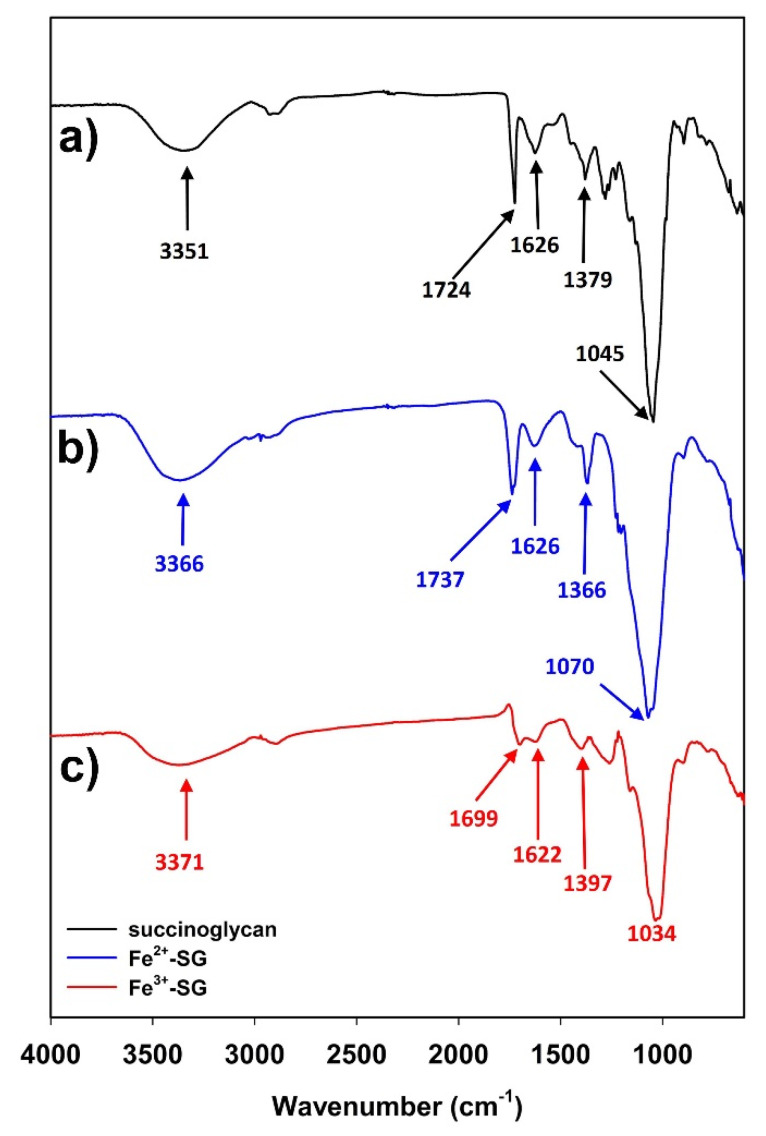
ATR-FTIR spectra of succinoglycan (**a**), Fe^2+^-SG sol (**b**), and Fe^3+^-SG hydrogel (**c**).

**Figure 4 polymers-12-00977-f004:**
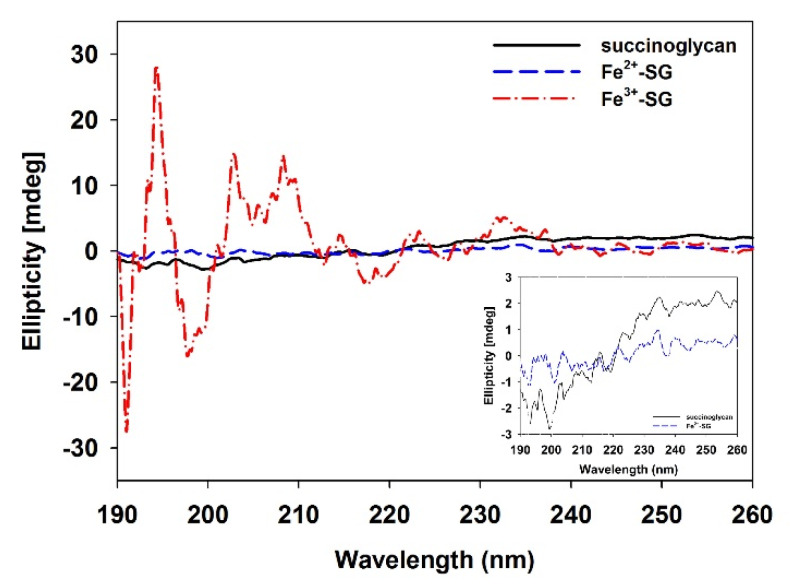
Circular dichroism (CD) spectra of the succinoglycan, succinoglycan mixed with the Fe^2+^ solution (Fe^2+^-SG), and Fe^3+^ solution (Fe^3+^-SG). The inset shows the spectra that details the succinoglycan and Fe^2+^-SG.

**Figure 5 polymers-12-00977-f005:**
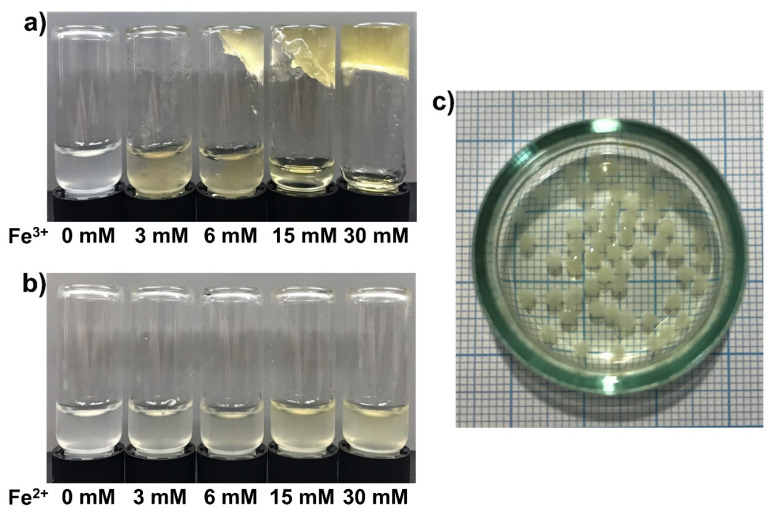
Results of the inverted vial tests. The Fe^3+^ (**a**) and Fe^2+^ (**b**) solution was mixed with the succinoglycan solution in vial, which was then turned upside down to check the gelation. The Fe ion concentrations were 0, 3, 6, 15, 30 mM (from left to right). Photograph of Fe^3+^-SG hydrogel beads (**c**).

**Figure 6 polymers-12-00977-f006:**
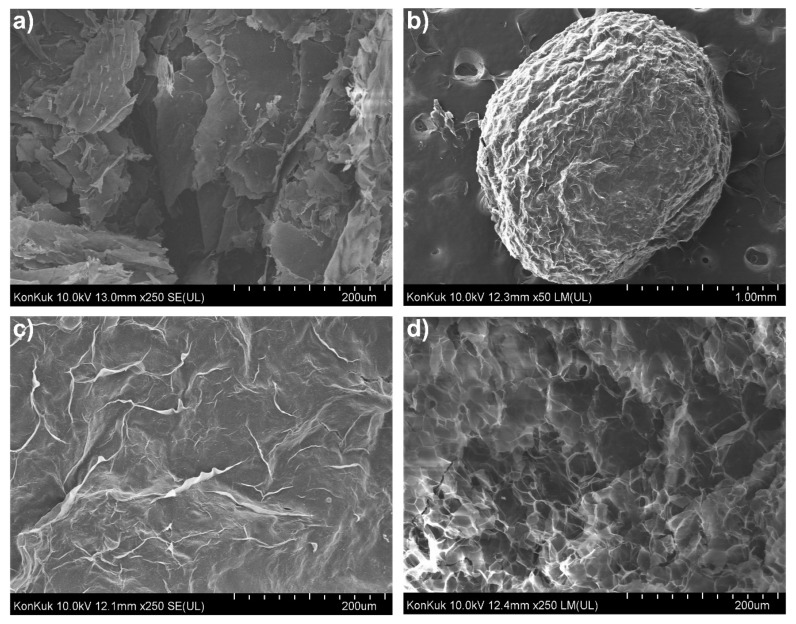
SEM images of succinoglycan (**a**), surface morphology of Fe^3+^-SG hydrogel beads in the dry state (**b**), dried Fe^3+^-SG hydrogel beads under 250× magnification (**c**), and the cross-sectional images of the Fe^3+^-SG hydrogel beads (**d**).

**Figure 7 polymers-12-00977-f007:**
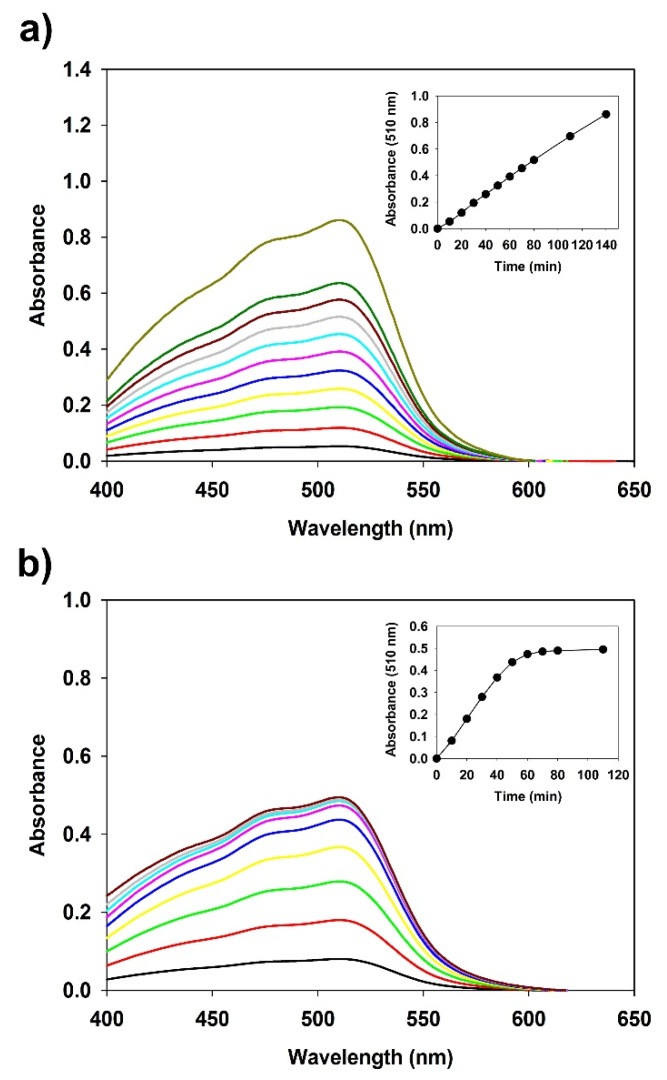
Changes in the absorbance spectra of Fe^2+^-phen complex in Fe^3+^-SG hydrogel beads. Beads with ascorbic acid (**a**) and beads with lactic acid under 405 nm visible light (**b**). Inset: Correlation of absorbance at 510 nm against time.

**Figure 8 polymers-12-00977-f008:**
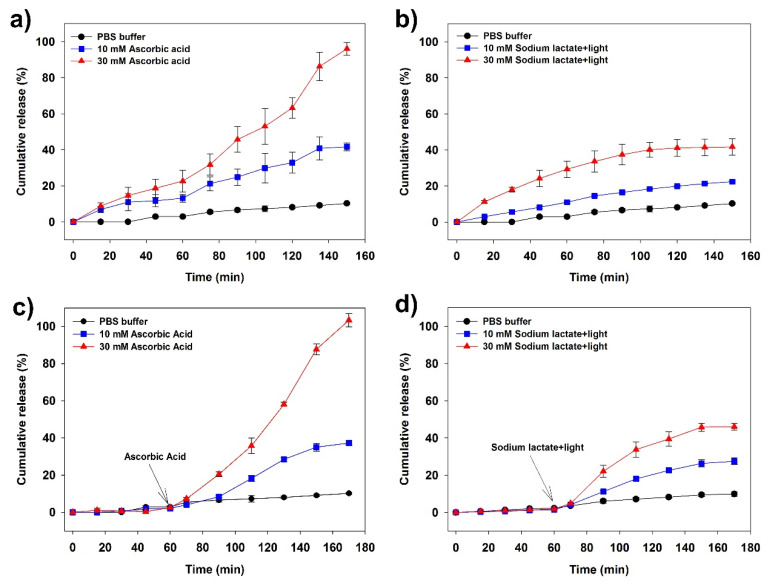
Congo red release percentage curves in response to ascorbic acid solution (**a**) and 405 nm visible light (**b**). In (**c**) and (**d**), release of Congo red from Fe^3+^-SG hydrogel beads during continuous exposure to a changing condition after 60 min at the presence of ascorbic acid or visible light.

**Figure 9 polymers-12-00977-f009:**
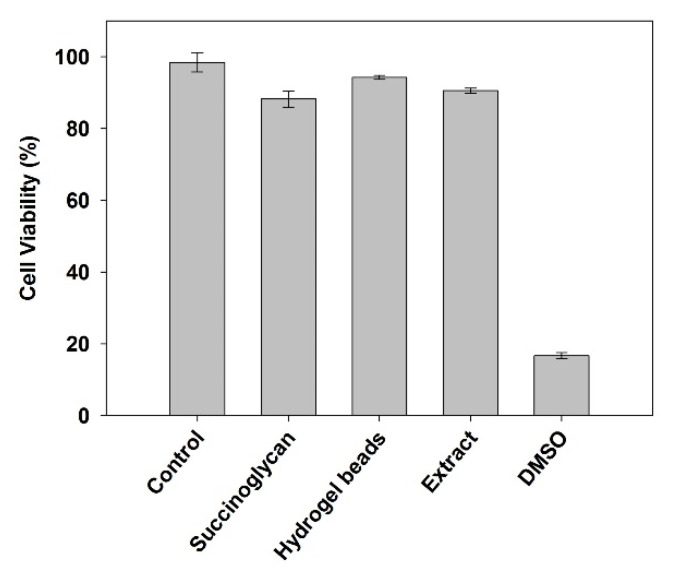
Human embryonic kidney 293 (HEK293) cell viability percentage following exposure to control, succinoglycan, dried Fe^3+^-SG hydrogel beads, extract, and DMSO.

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
