# Peer review of "Preparation of Succinoglycan Hydrogel Coordinated With Fe3+ Ions for Controlled Drug Delivery"

_polymers, 2020, doi:10.3390/polym12040977_

Round 1

Reviewer 1 Report

The paper presented describes the preparation of an iron / succinoglycan hydrogel, responsive to changes in the oxidation state of the coordinating iron species.  Good evidence is provided for hydrogel formation, and the mode of complexation is explored.  Evidence is also provided for the possible use of these materials as drug delivery vehicles; with the release of a model compound and evidence for biocompatibility.

Overall I think the paper is interesting and well presented.

I have a few comments to improve the text.

  • I found the explanation of the FTIR data confusing.  The data presented actually demonstrates the complexation well, and backs up the hypothesis given that bonding occurs through the C=O and -OH groups. However, the lists of wave numbers and shifts did not make sense.  For example Line 225 states "The peak at 1045.22 cm-1 was shifted to 1737.14 cm-1, 1366.28 cm-1, and 1070.15 cm-1" which seems to suggest that the peak has spilt, and this isn't the case.  I would recommend this section is revised to make the meaning clear.
  • On line 175 the abbreviation (GPC) in in the wrong place.
  • On line 373 the word used should be inserted between widely and as.

Reviewer 2 Report

In this article, the authors designed a succinoglycan hydrogel coordinated with Fe3+ ions for controlled drug delivery. Smart hydrogels will occupy a very important position in the field of controlled release delivery system. On the whole, the organization of article is reasonable. The English writing, logic and content are all appropriate and well conducted by the authors. However, a major modification is necessary before this article is accepted for publication on the “Polymers”. This suggestion is based on following reasons:

  • In Fig 2. The authors should discuss the concentration effect of Fe3+/Fe2+ on rheological behaviors.
  • Line 144, firstly, in congo red loading experiments, from my perspective, adding study on the encapsulation/loading efficiency of the congo red into the Fe3+-SG hydrogel could assess the drug delivery properties of the hydrogel. Secondly, the concentration range of ascorbic acid is 2 to 100 μg/mL in human plasma (Kim, Y.; Ha, N.; Kim, M.-G. Simultaneous Determination of l -Ascorbic Acid and Dehydroascorbic Acid in Human Plasma. Analytical Methods 2015, 7 (21), 9206–9210), but in this paper authors choose the concentration of ascorbic acid(5.28mg/mL) is far beyond the normal concentration in human plasma. Authors maybe should reconsider the concentration setting of reducing reagent. What’ more, it would be more perfect if authors could give an explanation why drug release experiment performed only in neutral pH condition.
  • Figure 5, in the photograph of the Fe3+-SG hydrogel beads, it’s better to add mean particle size value.
  • Figure 6, about the SEM images, in consideration of the reduction reaction process of ascorbic acid and Fe3+ is incomplete, and the optimal condition for this reduction reaction is acidic pH environment but in this paper pH is neutral, so theoretically, I think the hydrogel state will not completely change to sol state in the process of releasing. My suggestion is that authors can add the drug-loaded Fe3+-SG hydrogel complex SEM images in the before and after the release of the drug to intuitively reflect the morphological changes of the hydrogel complex carriers.
  • Figure 9, since there is a reduction process from Fe3+-SG to Fe2+-SG during the drug release process, the cytotoxicity of reduction products must be considered. Authors could supplement the cytotoxicity experiments of reduction products under the same conditions.

Reviewer 3 Report

This article reports Fe3+-coordinated succinoglycan (Fe3+-SG) hydrogel beads for stimuli-sensitive drug delivery. The hydrogel structure was thoroughly characterized by various techniques, such as CD, ATR-FTIR and SEM. Moreover, Fe3+-SG hydrogel beads showed controlled release of Congo red in response to a reducing agent and visible light. This manuscript is well organized and suitable to the scope of Polymers, and requires only minor revisions as follows:
1] In the statement “Low molecular weight succinoglycans can be chelated with Fe2+ to provide antioxidant activity through an anti-pentone reaction, thereby effectively controlling Fe biochemistry” (page 2, line 69), please replace “anti-pentone reaction” with “anti-Fenton reaction”.
2] There is no description on gel permeation chromatography (GPC) in the Materials and methods section, although it is an important information for characterization of succinoglycan. Please describe the method for GPC analysis, including the used column, solvent composition and molecular weight standards.
3] In Figure 8b, Fe3+-SG hydrogel beads showed faster release of Congo red with increasing the concentration of sodium lactate. But a different trend was observed in Figure 8d, which shows the addition of 10 mM sodium lactate induced faster release of Congo red than 30 mM sodium lactate. It would be more informative if the authors explain possible reasons for the discrepancy.
4] For Reference 31, journal name, volume number and page number are missing. Please correct this mistake.

Round 2

Reviewer 2 Report

The authors have made good modification about this paper. However, there are some errors to correct.

1) Eventhough the authors made the encapsulation efficiency test, they still didn't make the loading efficiency determination of the congo red. This is a key parameter for the drug carrier.

2) In Fig S5, the captain of one column is incorrect, which is Fe2+ not Fe3+.

3) The significant fugure of the ATR-IR is wrong due to the test resolution of 0.5 cm-1; Thus the presented wavenumber in the Section 3 and Fig 3 is wrong. fr example ,the right one should be 1700.5, 1701, 1701.5 .....  I strongly suggest the authors shoud correct significant fugures all through the paper.
